# Effects of Mud Content on the Setting Time and Mechanical Properties of Alkali-Activated Slag Mortar

**DOI:** 10.3390/ma16093355

**Published:** 2023-04-25

**Authors:** Shuaijun Li, Deyong Chen, Zhirong Jia, Yilin Li, Peiqing Li, Bin Yu

**Affiliations:** 1School of Civil Engineering and Geomatics, Shandong University of Technology, Zibo 255000, China; 2School of Transportation and Vehicle Engineering, Shandong University of Technology, Zibo 255000, China; 3Shandong Jiuqiang Group Co., Ltd., Zibo 255000, China

**Keywords:** alkali-activated slag, mud content, setting time, compressive strength, flexural strength

## Abstract

High mud content in the sand has a negative impact on cement mortar but there is little research on Alkali-activated slag (AAS) mortar. In order to explore the impacts of mud content in the sand on the performance of AAS mortar, this paper used sand that contains silt, clay, and a mixture of silt and clay; tested the setting time of AAS with different mud contents of 0%, 2%, 4%, 6%, 8%, and 10%; and measured the unconfined compressive strength and beam flexural strength of 3 d, 7 d, and 28 d AAS mortar specimens. The microstructure of AAS mortar with different kinds of mud was observed by scanning electron microscope (SEM), the elemental composition of the hydration product was tested by energy dispersive spectroscopy (EDS), and the AAS interaction mechanism with different kinds of mud was analyzed. The main conclusions are: the higher the mud content in the sand, the shorter the initial setting time and the longer the final setting time of AAS, mainly because the mud in the sand affects the hydration process; mud content above 4% causes a rapid decrease in the compressive and flexural strengths of AAS mortar, mainly because the mud affects the hydration process and hinders the bonding of the hydration product with the sand. When there is no mud in the sand, the main hydration product of AAS is dense calcium-alumina-silicate-hydrate (C-A-S-H) gel. When the sand contains silt, the hydration product of AAS is loose C-A-S-H gel. When the sand contains clay, the hydration products of AAS contain C-A-S-H gel and a small amount of sodium-aluminum-silicate-hydrate (N-A-S-H), and needle-like crystals. Loose gel and crystals have a negative effect on the AAS mortar strength.

## 1. Introduction

Ordinary Portland Cement (OPC) is the most widely used construction material because of its low cost, ease of pouring, good mechanical properties, durability, fire resistance, and wide availability [1,2]. However, the production process of cement emits large amounts of greenhouse gases and consumes large amounts of energy, limiting the sustainable development of concrete materials, and the climate crisis and environmental degradation are long-term challenges for humanity [3,4,5]. Alkali-activated slag (AAS) materials are a promising class of environmentally friendly cementitious materials, which are produced by the chemical reaction of slag with sodium silicate, sodium hydroxide, or another alkaline activator [6]. It has a high hardening rate, high early strength, corrosion and temperature resistance, and low production energy consumption [7,8,9,10,11]. Compared to OPC, AAS can significantly reduce the environmental impact of concrete products, emitting fewer greenhouse gases and consuming less energy and water due to the reuse of by-products from the steel industry [12]. Therefore, AAS is a more environmentally friendly alternative to OPC and has great potential [13,14,15].

Mud less than 75 μm is usually present on aggregate, which affects the performance of mortar and concrete [16,17]. They can prevent the cohesion between cementitious materials from forming weak interfacial layers, and they clump together to form weak areas; thus, they may have harmful effects on the properties of hardened concrete [17,18,19,20,21]. In addition, most mud, especially clay, with its large surface area and loose structure, has a great affinity for water and can also assimilate it into its microstructure, reducing concrete compatibility and increasing water consumption [22,23]. Hence, the mud attached to the aggregate has a non-negligible effect on the compatibility, mechanical properties, and durability of concrete [24,25,26,27,28], causing deterioration in the strength and freeze-thaw durability of concrete [17]. Mud content in sand adversely affects the strength of concrete [29]. Under normal conditions, the concrete slump decreases with the increase in mud content of aggregates [17,28,30,31], which will cause difficulties in the field construction of concrete. In addition, the variation in the properties of mortar and concrete depends on the chemical composition and mineral composition of the mud in the sand [16,32]. The sand–cement mixture with mud cannot be considered a simple mixture but a system in which the mud is combined with the hydrated cement through a secondary reaction. The exchange in cations and flocculation effects occur and the mud is still in a stable state when the mixture is in a freshly mixed state [31,32,33,34]. During the hardening of the cement mixture, the cement undergoes a hydration reaction to produce calcium silicate hydrate, calcium hydroxide, and hydrated aluminate, and the soluble alkali released during the hydration of calcium hydroxide and cement makes the pore solution strongly alkaline, triggering the erosion of the mud, leading to the decomposition of amorphous alumina and silica, which then combine with the calcium ions produced by the hydration reaction of the cement to produce secondary cementitious materials. This destroys the cement hydration products and reduces the mechanical properties of concrete [28,35,36,37,38,39,40].

Researchers have done a lot of work on the effect of mud content in the sand on cement and concrete but there is a lack of research on the influence of mud content on AAS mortar, and there are still many questions that need to be answered:(1)Does the mud affect the setting time of AAS, and is there the same effect for different mud contents and different kinds of mud;(2)Does the mud in the sand affect the compressive and flexural strengths of AAS mortar, and is there the same effect for different mud contents and different kinds of mud;(3)Is the threshold mud content of sand the same for AAS materials as the threshold mud content stipulated by the specification associated with the cement mixture?

In order to solve the above problems, this paper investigates the effects of silt, clay, and mixed mud at different contents on the setting time and mechanical properties of AAS mortar. SEM-EDS was used to observe the microstructure and hydration products of AAS mortar from different mud contents and to explain the reaction mechanism.

## 2. Materials and Methods

### 2.1. Materials

The slag was produced by Fuheng Mineral Products Trade Co., Ltd., Lingshou City, Hebei Province, China. The chemical composition analyzed by XRF is shown in Table 1.

The alkaline activator used was sodium silicate nine-hydrate (Na_2_SiO_3_-9H_2_O), analytically pure, white granules, supplied by Tianjin Dengfeng Chemical Plant (Tianjin, China).

The silt was taken from Gaoqing District, Zibo, Shandong Province, and the clay was taken from Zhangdian District, Zibo, Shandong Province, and filtered through a 75 μm sieve to remove the coarse particles. The chemical composition analyzed by XRF (BRUKER AXS GMBH, Beijing, China) is shown in Table 2. The particle size distribution, measured by Mastersizer 3000 Laser Diffraction Particle Size Analyzer (Malvern Instruments Ltd., Malvern, UK), is shown in Figure 1. The sand is ISO 679 standard sand (GSB08-1337) [41].

### 2.2. Testing Design

#### 2.2.1. Testing Grouping

To investigate the effect of mud content on the performance of AAS mortar, six different percentages of mud content—0%, 2%, 4%, 6%, 8%, and 10%—and three different types of mud—silt, clay, and mixed mud (silt and clay content were 1:1)—were used in this experiment. Activator percentage was determined to be 6% according to the *Technical standard for application of alkali-activated slag concrete* (JGJ/T 439-2018) [42]. The percentage of Na_2_O is 3%, accordingly, the modulus of sodium silicate is one. The proportions of the mixtures are summarized in Table 3. The dose of the activator is added as a percentage of precursor mass; mud content is added as a percentage of sand mass.

#### 2.2.2. Water–Binder Ratio

The best water consumption of mortar test was determined under the *Testing Methods of Cement and Concrete for Highway Engineering* (JTG 3420-2020) for standard consistency test, and the water–binder ratio of mortar test was fixed to 0.5 following the specification [43].

#### 2.2.3. Sample Maintaining

The maintenance of AAS mortar specimens followed the specification [43].

### 2.3. Test Methods

#### 2.3.1. Setting Time

According to the specification [43], after the standard consistency test, determine the water consumption, and pre-dissolve the solid activator in water in advance. Put the slag and solution in a mixing pot, stir slowly for 120 s, stop for 15 s, then quickly for 120 s. Used the Vicat apparatus to measure the setting time. When slag and activator are added into the water, it is the beginning of the AAS setting time; when the test needle is sunk to 4 mm ± 1 mm away from the bottom plate, it is the end of the initial setting time of AAS; when the ring attachment cannot leave traces on the specimen, it is the end of the final setting time of AAS. When approaching the initial setting time, it should be measured every 5 min (or less); when nearing the final setting time, it should be measured every 15 min (or less). It should be measured again immediately when reaching the initial or final setting, and the two conclusions are the same to determine the arrival of the initial or final setting time.

#### 2.3.2. AAS Mortar Strength

According to the specification [43] for the preparation of the AAS mortar specimen, the fresh mortar mixture is poured into the 40 mm × 40 mm × 160 mm triplex test mold two times, and each time it is poured, vibrated 60 times. Then, it is maintained under standard environmental conditions (temperature 20 °C ± 1 °C, relative humidity > 90%) for 24 h. After removal from the mold, the specimens are placed into the sink and maintained under standard conditions until reaching the age (3 d, 7 d, 28 d) for testing. The compressive and flexural strength tests of the cementitious sand are measured using cementitious sand compressive and flexural apparatus. The average values of the three specimens are determined by the flexural and compressive strength of the AAS mortar for the corresponding curing age.

#### 2.3.3. Micro-Analysis

Crushed specimens were kept in anhydrous ethanol for 7 d to stop the hydration reaction after the flexural and compressive strength tests [44,45]. Then, the specimens were removed from the anhydrous ethanol and dried to an absolutely dry state. The morphology and elemental composition of the hydration products were determined using SEM and EDS to determine the hydration products of the AAS mortar and the reaction mechanism. The QUANTAFEG 250 field emission scanning electron microscope was manufactured by FEI, Hillsboro, OR, USA.

## 3. Test Results and Analysis

The experimental results of the different-parameter samples are shown in Table 4.

### 3.1. Setting Time

Figure 2 shows the variation of AAS setting time for the different mud and content. The initial setting time of the AAS gelling material gradually decreases with the rise in the silt content, from 46 min to 28 min, which is 39% shorter. In contrast, the final setting time gradually increases with the rise in the silt content, from 150 min to 225 min, which is 50% longer.

The addition of clay and mixed mud shows the same trend as the addition of silt but the change in setting time is more significant because the surface area of clay is larger, which made its adsorption stronger: the initial setting time of AAS gelling material reduces from 46 min to 17 min, shortened by 63%, and the final setting time extends from 150 min to 244 min, extended by 62.6%.

The initial setting time of AAS with the addition of mixed mud reduces from 46 min to 23 min, shortened by 50%, and the final setting time extends from 150 min to 238 min, increased by 58.6%.

The above phenomenon is mainly due to the water absorption of the mud. With the increase in the mud content, the water absorption of the mud increases, which indirectly reduces the water-binder ratio (w/b) in the early stage. The setting time of AAS depends on the OH^−^ and Ca^2+^ ion concentration. Lower w/b increases the OH^−^ ion concentration. Consequently, the rate of breaking the Ca-O, Si-O-Si, and Al-O-Al bonds is sped up, which leads to the increase in Ca^2+^ ion concentration and accelerates the formation of hydration products and the arrival of the initial setting state [20,46,47]. However, the water absorption of mud also delays the hydration process, especially with high kaolinite content [48,49,50,51], leading to a longer final setting time. As for the effect of the particle sizes of mud on setting time, the capability of liquid absorption of clay with a smaller particle size is stronger than that of silt with a larger particle size [46]. Therefore, the initial setting time of AAS containing clay is shorter, and the final setting time is longer.

### 3.2. Mechanical Property

Figure 3 shows that the flexural and compressive strengths of all specimens increased with curing age. Figure 3a shows the effect of different silt content on the compressive strength of AAS mortar at 3 d, 7 d, and 28 d. Figure 3a shows that the compressive strength of AAS mortar has no significant effect when the silt content is 2%, compared with the specimen without silt (0% mud content); when the silt content is 4%, the 3 d, 7 d, and 28 d compressive strength of AAS mortar decreases slightly, with a decline of 2.8~7.2%; when the silt content increases to 6~10%, the compressive strength of AAS mortar decreases rapidly at 3 d, 7 d, and 28 d, reaching a minimum of 10%, with a decrease of 25.9~30.4%.

Figure 3b shows the effect of different silt content on the flexural strength at 3 d, 7 d, and 28 d. Similar to the compressive strength, the silt has a harmful effect on the flexural strength of the AAS mortar. Compressive strength gradually decreases with rising silt content in the sand during all the curing ages; decreases by 6.3~7.7% when the silt content increases from 0% to 4%; and the flexural strength decreases by 17.3~17.6% (27.5~33.3%) when the silt content increases to 6% (10%). The greater content of silt, the more significant the adverse effect on the mechanical properties of AAS mortar.

Figure 4a shows the effect of different clay content on the compressive strength of AAS mortar at 3 d, 7 d, and 28 d. The compressive strength of AAS mortar at different curing ages shows a trend of slight increase and then decrease with the rise of clay content, and its strength reaches the peak at about 2% of clay content, and the peak does not change with the curing age, and drops to the lowest value at 10% of clay content. The compressive strength of AAS mortar increases by 1.1~4.8% when the clay content is 2%; the strength decreases by 2.8~3% when the content reaches 4%; the strength decreases by 20~21.4% when the content is 6%; the strength decreases by 27.3~29.1% when the content is 8%; and the strength decreases by 35.3~35.7% when the content is 10%.

Figure 4b shows the effect of different clay content on the flexural strength of AAS mortar at 3 d, 7 d, and 28 d. Similar to the compressive strength, a small amount of clay has no effect on the flexural strength of the mortar but when the clay content is above 4%, clay has a harmful impact on the flexural strength. Figure 4b shows that the flexural strength increases from 4.5 to 12.3% when the clay content increases from 0 to 2% at all curing ages, and the flexural strength decreases by 20.6~23.5% (35.2~39.7%) when the clay content increases to 6% (10%).

Figure 5 shows the effect of different mixed mud content on the compressive and flexural strengths of AAS mortar at 3 d, 7 d, and 28 d. The compressive and flexural strengths increase by 0.4~0.5% and 1.2~2.2%, respectively, when the mixed mud content is 2%; the compressive and flexural strengths decrease by 3.4~7.6% and 6.3~9.9%, respectively, when the content is 4%; the compressive and flexural strengths reduce by 17.7~22% (28.4~34.4%) and 14.3~22.2% (34.9~37.4%), respectively when the content reaches 6% (10%).

Figure 6 visually shows the changes in the compressive and flexural strengths of AAS mortar resulting from the amounts of clay, silt, and mixed mud. The clay has a more significant effect on the compressive and flexural strengths of AAS mortar compared to the silt and mixed mud. When the mud content is less than 4%, the effect of mud is not significant on the compressive and flexural strengths of AAS mortar; when the content is more than 4%, mud leads to a rapid decrease in the compressive and flexural strengths of AAS mortar.

Overall, the mud in the sand has a non-negligible effect on the compressive strength and flexural strength of AAS mortar. When the silt and clay content is less than 4%, the mud has almost no effect on the compressive and flexural strengths of AAS mortar, which is because the mud fills the pores of the sand skeleton [52] and reduces the negative effect caused by the addition of mud. When the mud content exceeds 4%, the compressive and flexural strengths of AAS mortar decrease rapidly, mainly for the following reasons: (1) mud absorbs water from the fresh mortar and this affects the hydration process [49,50,51]; (2) part of the mud wraps around the sand particles, which interferes with the bonding of sand with cementitious materials [52]; (3) as the mud content increases, this results in a decrease in overall strength because the overall strength of the specimen is shared by hydration products, sand, and mud, while the strength of mud is lower than that of hydration products and sand [53,54]; and (4) the active components in clay minerals will also be activated by alkaline and generate additional cementitious products [55]. Clay is finer than silt, and with more active components, it has different effects on the compressive and flexural strengths of AAS mortar [56].

### 3.3. Mechanism Analysis

Figure 7 shows the microstructure and EDS diagrams of AAS hydration products under different conditions. Figure 7a–c represents the microstructure of 28 d AAS mortar with 0% mud content, 6% silt content, and 6% clay content, respectively. Figure 7d–f represents the EDS diagrams of the corresponding points in a–c, respectively. It can be observed in Figure 7a that when the AAS mortar does not contain mud, the alkali slag product is structurally dense, with more gel and better integrity. The EDS results in Figure 7d clearly show the elemental composition of the corresponding points of the A0 specimen, and there are a large number of alkali metal ions in the gel products, among which Ca^2+^ and Al^3+^ are dominant, almost without Na^+^, mainly because compared with Na^+^, the Ca^2+^, and Al^3+^ are more attracted to negative charges and will preferentially combine with anions [57,58,59,60], which makes the main reaction products of the slag C-S-H and C-A-S-H gels, they are the primary source of the strength of the AAS mortar.

It can be observed in Figure 7a,b that the specimen of A0 has a better degree of hydration reaction. In contrast, S6 has a slower hydration reaction process. It produces a loose flocculent gel, which is the reason for the decrease in compressive and flexural strengths of the AAS mortar-containing silt. The EDS results in Figure 7e clearly show the elemental composition of the corresponding points of the S6 specimen. Compared to Figure 7d, Al elements cannot be found, which is mainly because the silt slows down the hydration process, resulting in the reduction of C-A-S-H gels, forming loose flocculent gels. The finding is similar to the phenomenon of alkali-activated fly ash with silt studied by Sheng et al. [61].

Figure 7c shows that a large amount of square or needle-like products appear in the surface layer, which is similar to the phenomenon previously shown by Karozou et al. using alkali activators to activate clay [62]. It can be seen by EDS analysis in Figure 7f that there is a significant increase in silicon elements in the square or needle-like products and a small amount of Na^+^. The main reason is that the slag material with high calcium will dissociate a large number of free Ca^2+^ during the alkaline being activated, and the clay raw material component contains a large amount of silicon and aluminum elements, which will produce a large amount of [AlO_4_]^5−^ and [SiO_4_]^4−^ when activated and polymerize with alkali metal Ca^2+^ to generate C-(A)-S-H gel. Due to the decrease in the calcium-silica ratio, some [AlO_4_]^5−^ and [SiO_4_]^4−^ reacted with Na^+^ to form a small amount of N-A-S-H gel [63,64]. That is similar to the results of Yonghua et al. using slag and fly ash to solidify clay [53]. The N-A-S-H gel exhibits characteristics common to both amorphous and pseudo-zeolitic or proto-zeolitic structures [65,66], destroying the original cemented dense gel structure, and reducing the AAS mortar strength.

## 4. Conclusions

Based on this study on the effect of different contents of silt, clay, and a mixture of silt and clay on the setting time, compressive strength, and flexural strength of AAS mortar, the major conclusions are drawn as follows:(1)Mud in the sand significantly affects the setting time of AAS. As the mud absorbs water, the alkali concentration increases, which speeds up the reaction of AAS and shortens the initial setting time of AAS but the mud slows down the hydration process and makes the final setting time longer. With the increase in mud content, this phenomenon also becomes more and more significant. Clay has a more significant influence than silt;(2)With the increase in the silt content in the sand, the compressive strength and flexural strength of AAS mortar both decrease. When the silt content is less than 4%, there is no significant effect of silt on the mechanical properties of AAS mortar. When the silt content exceeds 4%, the silt affects the hydration process of AAS that leads to the weakening of the bond between the gel and the sand, producing loose flocculent gel and reducing the mechanical properties of AAS mortar significantly;(3)The clay in the sand has almost no negative impact on the compressive and flexural strengths of AAS mortar when the clay content is less than 4%; the strength of AAS mortar decreases rapidly when the content exceeds 4%. The clay slows down the hydration process of AAS, hinders the bonding between the gel and sand, and the clay minerals destroy the original dense gel structure. Microscopic analysis shows that a small amount of N-A-S-H gel and partially ordered crystals are produced, which destroys the dense gel structure;(4)The recommended threshold mud content of sand for AAS mortar is 4% according to the setting time and strength. However, other properties of AAS mortar with mud should be studied further in order to calibrate the recommended thresholds.

## Figures and Tables

**Figure 1 materials-16-03355-f001:**
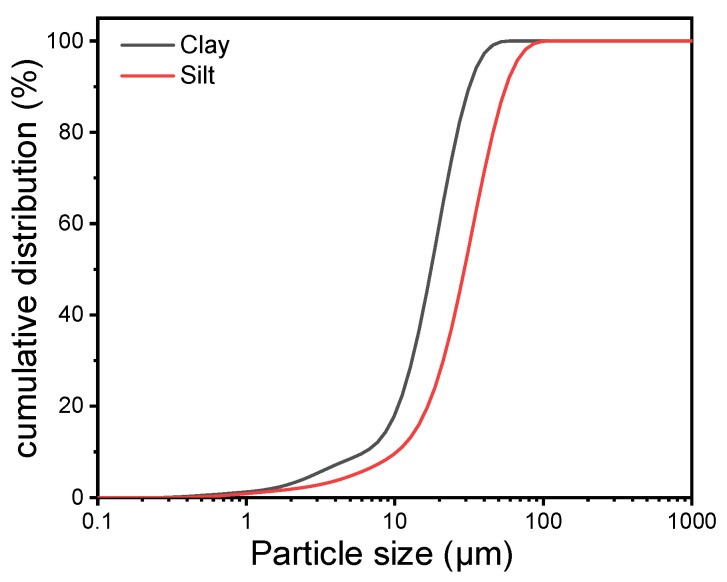
Particle size distribution of silt and clay.

**Figure 2 materials-16-03355-f002:**
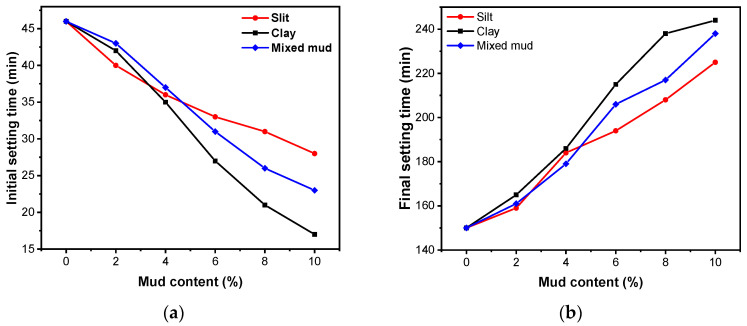
Setting time of AAS with different mud content: (**a**) Initial setting time; (**b**) Final setting time.

**Figure 3 materials-16-03355-f003:**
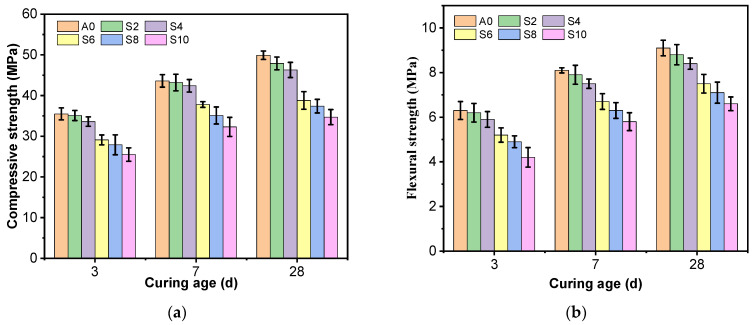
Compressive and flexural strengths of AAS mortar with different silt content: (**a**) Compressive strength; (**b**) Flexural strength.

**Figure 4 materials-16-03355-f004:**
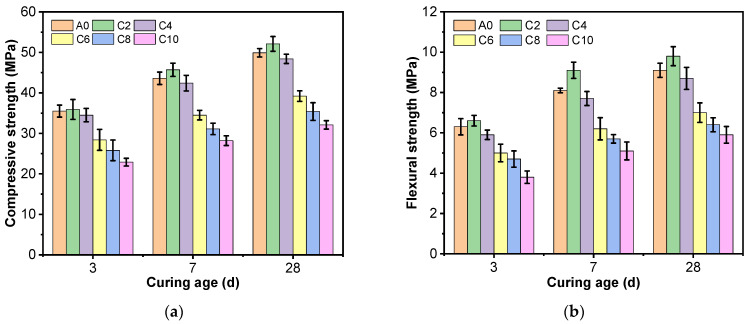
Compressive and flexural strengths of AAS mortar with different clay content: (**a**) Compressive strength; (**b**) Flexural strength.

**Figure 5 materials-16-03355-f005:**
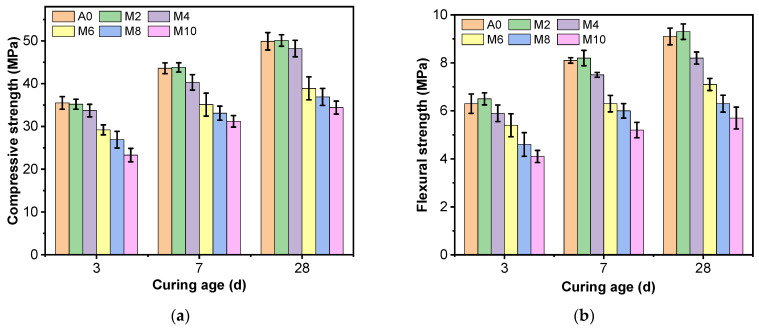
Compressive and flexural strengths of AAS mortar with different mixed mud content: (**a**) Compressive strength; (**b**) Flexural strength.

**Figure 6 materials-16-03355-f006:**
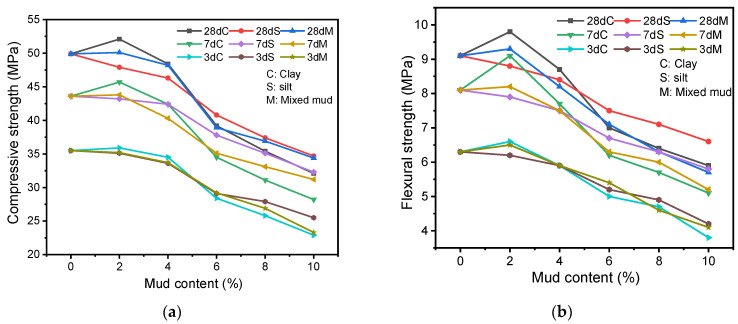
Compressive and flexural strengths of AAS mortar with different mud content: (**a**) Compressive strength; (**b**) Flexural strength.

**Figure 7 materials-16-03355-f007:**
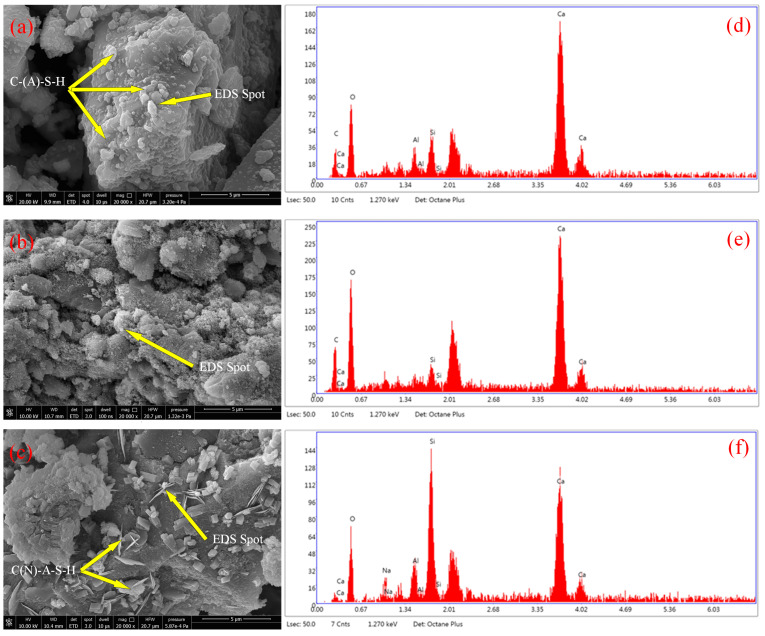
SEM and EDS diagrams of AAS mortar cured 28 d: (**a**) A0; (**b**) S6; (**c**) C6; (**d**) EDS image of A0; (**e**) EDS image of S6; (**f**) EDS image of S6.

**Table 1 materials-16-03355-t001:** Chemical composition of slag.

Oxide	CaO	Al_2_O_3_	SiO_2_	MgO	Fe_2_O_3_	TiO_2_	SO_3_	MnO	Na_2_O
Mass (wt.%)	42.6	14.2	27.8	8.09	0.378	1.2	2.46	0.401	0.55

**Table 2 materials-16-03355-t002:** Chemical composition of clay and silt.

Chemical Composition	SiO_2_	Al_2_O_3_	Fe_2_O_3_	K_2_O	Na_2_O	MgO	SO_3_	CaO	TiO_2_	MnO	P_2_O_5_	LOI
Clay (wt.%)	57.6	33.6	1.1	0.3	0.4	0.02	—	1	0.04	0.02	0.03	4.94
Silt (wt.%)	63.7	15.9	3.07	2.6	1.82	1.87	0.06	5.72	0.48	0.05	0.13	3.8

**Table 3 materials-16-03355-t003:** Testing Groups.

Mix Id	Activator Percentages	Na_2_O Percentages	Mud Percentages
A0	6%	3%	0
S2	6%	3%	Silt-2%
S4	6%	3%	Silt-4%
S6	6%	3%	Silt-6%
S8	6%	3%	Silt-8%
S10	6%	3%	Silt-10%
C2	6%	3%	Clay-2%
C4	6%	3%	Clay-4%
C6	6%	3%	Clay-6%
C8	6%	3%	Clay-8%
C10	6%	3%	Clay-10%
M2	6%	3%	Mixed mud-2%
M4	6%	3%	Mixed mud-4%
M6	6%	3%	Mixed mud-6%
M8	6%	3%	Mixed mud-8%
M10	6%	3%	Mixed mud-10%

**Table 4 materials-16-03355-t004:** Testing results of different-parameter samples.

Mix Id	Initial Setting Time (Min)	Final Setting Time (Min)	Compressive Strength (MPa)			Flexural Strength (MPa)		
			3 d	7 d	28 d	3 d	7 d	28 d
A0	46	150	35.5	43.6	49.9	6.3	8.1	9.1
S2	40	159	35.1	43.2	47.9	6.2	7.9	8.8
S4	36	184	33.6	42.4	46.3	5.9	7.5	8.4
S6	33	194	29.1	37.8	40.8	5.2	6.7	7.5
S8	31	208	27.9	35.1	37.4	4.9	6.3	7.1
S10	28	225	25.5	32.3	34.7	4.2	5.8	6.6
C2	42	165	35.9	45.7	52.1	6.6	9.1	9.8
C4	35	186	34.5	42.4	48.4	5.9	7.7	8.7
C6	27	215	28.4	34.5	39.2	5	6.2	7
C8	21	238	25.8	31.1	35.4	4.7	5.7	6.4
C10	17	244	22.9	28.2	32.1	3.8	5.1	5.9
M2	43	161	35.2	43.8	50.1	6.5	8.2	9.3
M4	37	179	33.7	40.3	48.2	5.9	7.5	8.2
M6	31	206	29.2	35.1	38.9	5.4	6.3	7.1
M8	26	217	26.9	33.1	36.9	4.6	6	6.3
M10	23	238	23.3	31.2	34.4	4.1	5.2	5.7

## Data Availability

The data presented in this study are available on request from the corresponding author. The data are not publicly available due to the need for confidentiality.

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
