# Peer review of "Effects of Mud Content on the Setting Time and Mechanical Properties of Alkali-Activated Slag Mortar"

_materials, 2023, doi:10.3390/ma16093355_

Round 1

Reviewer 1 Report

Review report on

Effects of mud content on the Setting Time and Mechanical Properties of Alkali-activated Slag Mortar  

The paper is well-organized and includes new contributions with good merits for publication. I humbly ask the authors to carefully read the attached concerns and make major modifications to enhance the presentation of their paper.

- What is the main difference between the properties of different raw materials used in this work?

- What is the main objective behind the current study? It is beneficial for the readers to add more explanations about the novel contribution of this method from theoretical/experimental viewpoints.

- The authors must explicitly declare the assumptions and limitations of their model. It seems it's application is quite restricted.

- The literature review is inadequate and one-sided. There are contributions about “Mortars and Concrete” which should be mentioned. The authors should appropriately extend this section by discussing more relevant works focusing on different methods and models in the literature. For example, it is suggested to read and discuss about the following relevant works:

- Saberi Varzaneh, A., Naderi, M. Experimental and Finite Element Study to Determine the Mechanical Properties and Bond Between Repair Mortars and Concrete Substrates. Journal of Applied and Computational Mechanics, 2022; 8(2): 493-509.

and other related works.

- It is suggested to add more in-depth explanation of the model, its justification and more discussions on the results.

- The paper should be carefully double-checked from grammatical point of view.

Reviewer 2 Report

Dear authors,

In the manuscript, Effects of mud content on the Setting Time and Mechanical Properties of Alkali-activated Slag Mortar, the authors have done the effect of mud in the sand used for Alkali-activated slag mortar. In order to solve the problems the next is investigated: the effect of the setting time of Alkali-activated slag, and does the effect is the same for different mud; whether the mud in the sand reacts with the Alkali-activated slag mortar and whether the influence of different mud is the same and what is the reaction mechanism. The researchers investigate the effects of silt, clay, and mixed mud at different content on the setting time and mechanical properties-compressive strength, and flexural strength of Alkali-activated slag mortar. The authors have the next conclusion: The mud in the sand has a significant effect on the setting time of Alkali-activated slag. With the increase of the silt content in the sand, the compressive strength and flexural strength of Alkali-activated slag mortar all decrease. The effect of clay in the sand is complex on the compressive and flexural strength of Alkali-activated slag mortar. This study recommends that the mud content of sand for Alkali-activated slag mortar should not exceed 4%.

I consider that the research is presented very concisely and in a simple, accessible way. The authors were very specific and clear about the methodology, tests, results, and discussion.  All of these is the reason that I recommend the manuscript be accepted in this form.

Sincerely

Author Response

Thank you for your valuable suggestions!

Reviewer 3 Report

The theme of the manuscript is interesting and relevant. However, the manuscript has several flaws that need to be corrected for publication to be considered. Basically, the discussion of the results needs to be completely revised and complemented. The authors do not compare the results obtained with results from the literature and merely explain the observed behaviors. Some specific points are highlighted:

1)      “The main conclusions are: mud in the sand has a significant effect on the setting time, compressive strength, and flexural strength of AAS mortar, especially the setting time.” Too generic sentence for the abstract. To remove.

2)      “The higher the mud content in sand, the shorter the initial setting time and the longer the final setting time of AAS.” Why does it happen?

3)      “Therefore, 43 AAS is a more environmentally friendly alternative to OPC and has the great potential 44 [12-14].” Check existing literature on the environmental impact of Alkali-activated slag (AAS) mortar. The activator solution used emits large amounts of CO2 as well. Review the sentence.

4)      “Researchers have done a lot of work on the effect of mud in the sand on cement, but 69 there are still many problems with the effect of mud in the sand used for AAS mortar, as 70 shown below:” Highlight the main studies on the effect of mud on the properties of Alkali-activated slag (AAS) mortar.

5)      “It is hoped to 81 determine whether and how the mud affects AAS mortar and the threshold of AAS mortar mud content.” Remove the sentence from the introduction.

6)      Remove Fig. 1.

7)      “The sand is ISO standard sand” Quote ISO number.

8)      Enter the granulometric distribution of the materials used.

9)      How were activator and Na2O contents defined?

10)   “Use the Vicat apparatus to measure the setting time. When approaching the initial setting time, it should be measured every 5 minutes (or less), when nearing the final setting time, it should be measured every 15 minutes (or less). When reaching the initial or final setting, it should be measured again immediately, and the two conclusions are the same to determine the arrival of the initial or final setting time.” What values ​​indicate the beginning and end of Alkali-activated slag (AAS) mortar setting time?

11)   “Crushed specimens were kept in anhydrous ethanol for 7d to stop the hydration re-142 action after the flexural and compressive strength tests. Then, the specimens were re-143 moved from the anhydrous ethanol and dried to an absolutely dry state” Why was ethanol used? Usually, isopropyl alcohol or acetone is used.

12)   Figure 2: First, discuss the setting time of the Alkali-activated slag (AAS) mortar produced concerning the values ​​reported in the literature.

13)   When discussing the setting time results, the authors did not cite a reference to justify or compare the results.

14)   “It can be seen from Fig. 3(a) that the 3d, 7d, and 28d compressive strength of AAS mortar decreased slightly when the silt content was 4%, with compressive strength decreasing by 2.8% to 7.2%, compared with specimen without silt (0% mud content); When the silt content increases to 6%~10%, the compressive strength of AAS mortar decreases rapidly at 3, 7, and 28 days, reaching a minimum of 10%, with a decrease of 25.9%~30.4%. It is necessary to perform a statistical analysis to verify which values ​​have statistically significant differences. For instance, the compressive strength of A0 and S2 are very close, possibly without statistical differences. This type of analysis needs to be incorporated into the discussion of results.

15)   Suggestion: present the compressive strength values ​​of AAS mortar with silt and clay together. Same thing for bending strength.

16)   “In summary, the mud in the sand has a non-negligible effect on the compressive strength and flexural strength of AAS mortar. The mud will weaken the compressive strength and flexural strength of AAS mortar, but a small amount of clay can slightly improve the compressive strength and flexural strength. The more mud in the sand, the more significant the effect of mud content on the compressive and flexural strength.” Very simplistic discussion for a scientific manuscript. It is essential to deepen the discussion, presenting similar results from the literature.

17)   Rewrite the discussion of SEM and EDS results. The technique makes a very punctual analysis. In this way, it is very difficult to obtain conclusive findings with a purely qualitative analysis.

Round 2

Reviewer 1 Report

The revised version is acceptable for publication.

Author Response

(The authors gave the same response as above.)

Reviewer 3 Report

Accept 

Author Response

(The authors gave the same response as above.)
